# SntB Affects Growth to Regulate Infecting Potential in *Penicillium italicum*

**DOI:** 10.3390/jof10060368

**Published:** 2024-05-21

**Authors:** Chunyan Li, Shuzhen Yang, Meihong Zhang, Yanting Yang, Zhengzheng Li, Litao Peng

**Affiliations:** College of Food Science and Technology, Huazhong Agricultural University, Wuhan 430070, China; 2021309110048@webmail.hzau.edu.cn (C.L.); yszhen@mail.hzau.edu.cn (S.Y.);

**Keywords:** *Penicillium italicum*, gene knockout, *SntB*, carbon starvation, RNA sequence

## Abstract

*Penicillium italicum*, a major postharvest pathogen, causes blue mold rot in citrus fruits through the deployment of various virulence factors. Recent studies highlight the role of the epigenetic reader, *SntB*, in modulating the pathogenicity of phytopathogenic fungi. Our research revealed that the deletion of the *SntB* gene in *P. italicum* led to significant phenotypic alterations, including delayed mycelial growth, reduced spore production, and decreased utilization of sucrose. Additionally, the mutant strain exhibited increased sensitivity to pH fluctuations and elevated iron and calcium ion stress, culminating in reduced virulence on Gannan Novel oranges. Ultrastructural analyses disclosed notable disruptions in cell membrane integrity, disorganization within the cellular matrix, and signs of autophagy. Transcriptomic data further indicated a pronounced upregulation of hydrolytic enzymes, oxidoreductases, and transport proteins, suggesting a heightened energy demand. The observed phenomena were consistent with a carbon starvation response potentially triggering apoptotic pathways, including iron-dependent cell death. These findings collectively underscored the pivotal role of *SntB* in maintaining the pathogenic traits of *P. italicum*, proposing that targeting *PiSntB* could offer a new avenue for controlling citrus fungal infections and subsequent fruit decay.

## 1. Introduction

Citrus fruits are susceptible to pathogenic fungal infections throughout the whole postharvest period [1]. Among the various postharvest diseases affecting citrus, green and blue mold are regarded as important industrial challenges [2,3]. Notably, blue mold, caused by *Penicillium italicum*, progresses more gradually but exhibits substantial resistance to cold conditions and low water availability and has a propensity for rapid spread and contamination of numerous healthy oranges [4,5,6]. This can lead to considerable economic damages. Currently, chemical fungicides such as imazalil, prochloraz, and triazolone constitute the primary strategy against blue mold in citrus fruits [7]. However, the extensive use of these fungicides has led to risks of pesticide residues, environmental contamination, and the emergence of resistant pathogen strains [3,8]. The molecular mechanisms underlying *P. italicum* invasion of citrus fruits remain insufficiently elucidated. Therefore, unraveling the molecular pathways of *P. italicum* invasion on citrus fruits is of paramount importance for innovating novel antimicrobial compounds and improving biocontrol strategies.

Virulence factors are pivotal in facilitating pathogen colonization within hosts, serving as key components in the establishment of infections [9,10]. The histone modification protein SntB, identified as an epigenetic reader, has been implicated as an important virulence factor in several fungi, which influenced a range of biological processes including fungal development, secondary metabolite synthesis, and virulence. The deletion of *SntB* increased global levels of H3K9K14 acetylation and reduced sclerotia formation, and heterokaryon compatibility and the ability to colonize host corn seeds were impaired in *Aspergillus flavus* [11]. Moreover, *SntB* has been shown to play a critical role in regulating autophagy. Deletion mutants of *Snt2* exhibited reduced sporulation and biomass accumulation of *Fusarium oxysporum*, along with upregulation of autophagy-related genes [12]. *SntB* could modulate the virulence of *Penicillium expansum* toward apples by affecting the production of patulin and citrinin [13]. Similar findings were observed in *Neurospora crassa* and *Magnaporthe oryzae*, which underscores the importance of *SntB* in fungal adaptation and stress response [14,15].

Other notable virulence factors include *LaeA*, *PacC*, and *CreA*. *LaeA* is regarded as the secondary metabolism global regulator since it coordinates the expression of various secondary metabolite clusters [16,17,18]. In *P. expansum*, *LaeA* regulates the biosynthesis of patulin, which impacts virulence related to this pathogen and is mediated by sucrose [19]. The pH regulatory factor *PacC* promotes pathogen colonization by generating organic acids to reduce the host pH at infection sites [20,21]. In both *P. expansum* and *Aspergillus carbonarius*, *PacC* directly regulates glucose oxidase (GOX), which catalyzes the oxidation of glucose to gluconic acid, facilitating host colonization [22,23]. In *Fusarium fujikuroi*, *ΔpacC* mutants did not exhibit expression of *FUB1*, the key polyketide synthase gene in the fusaric acid gene cluster [24,25]; *CreA* is the carbon metabolism repression regulator and regulates carbon source utilization by modulating carbon catabolite repression (CCR) activity in filamentous fungi [26]. Recent studies have discovered that *CreA* regulates growth and pathogenicity in *Valsa mali* and that *MoCreA* is essential for asexual development and pathogenicity in *Magnaporthe oryzae* [27,28]. Notably, *SntB* has been reported to positively regulate these global virulence and secondary metabolism regulators [13], suggesting its involvement in multiple critical biological processes and metabolic pathways.

The principal aim of this investigation was to elucidate the gene function and expression dynamics of *SntB* in *P. italicum* via Agrobacterium-mediated gene disruption complemented by transcriptomic analysis and to unveil novel mechanisms underlying the biology of citrus pathogens.

## 2. Materials and Methods

### 2.1. Fungal Strains and Plasmids

The *P. italicum* strain, isolated from navel orange fruit (*Citrus sinensis* L. Osbeck) and displaying symptoms of blue mold, was identified through morphological assessment of the colonies in our previous study [29]. To enhance the efficiency of homologous recombination, we previously developed a Ku70-deficient strain from *P. italicum*; this strain was employed as the parental control for all further experimental activities.

*Escherichia coli* DH5α and *Agrobacterium rhizogenes* AGL1 were employed for plasmid construction and storage from Shanghai Weidi Biotechnology (Shanghai, China). Plasmid pCAMBIA3300, commonly utilized for fungal transformation [30], and plasmid pTFCM containing the hygromycin resistance gene for subsequent fragment amplification were stored in our laboratory.

### 2.2. Construction and Verifification of Knockout Mutant

The orthologous gene sequence of *P. italicum SntB* was identified in GenBank (JQGA01000471.1) by using the Local BLAST method with the gene sequence of *A. flavus SntB* (AFLA_029990) against the *P. italicum* genome scaffolds, pinpointing its location at PITC_081300. For *PiSntB* knockout, flanking regions of the 5′-region (1383 bp) and 3′-region (1549 bp) of the *PiSntB* gene were PCR amplified from the genomic DNA of *PiKu70*. Meanwhile, the hygromycin-resistant fragment was used as a selective marker, which was from plasmid pTFCM (Appendix A). The vector pcmbia3300 was linearized using the restriction endonuclease XbaI, BamHI, after which the fragments were cohesively ligated utilizing a one-step cloning kit (Vazyme Biotechnology Co., Ltd., Nanjing, China).

The ligation reaction was later transformed into DH5α competent cells, performed according to the manufacturer’s instructions, and successfully ligated plasmid was verified using gene-specific primers (Appendix A), resulting in the creation of the knockout vector. The plasmid containing the knockout cassette was transferred into *Agrobacterium* AGL1 to generate knockout mutant strains, as prescribed by the method developed by Zhao [31]. Additionally, the obtained transformants were screened for resistance using the 50 µg/mL hygromycin B, and PCR identification was carried out employing specific detection primers to confirm the deletion of the *SntB* gene.

### 2.3. Physiological Analysis in P. italicum

#### 2.3.1. Radial Growth, Spore Production, and Spore Germination

Radial growth was examined according to the protocol described by Yang et al. [29]. Fungal spores from the same number of days of incubation were washed down with sterile distilled water, adjusted to a concentration of 1 × 10^6^ spores of each strain, and incubated in Potato Sucrose Agar (PSA) medium for 36 h. After incubation, a fungal disc was made with a 5 mm punch and back-pasted into fresh PSA medium, and the radial growth of the cultures was measured daily. The disc incubated for 48 h was washed as completely as possible with 1 mL sterile water, and the spore production was calculated. Based on the method of Wang et al. [32], 0.5% agar-PSA medium was mounted on a sterile glass slide (2.5 × 7.5 cm), and 20 μL suspension was inoculated on it and cultured at 26 °C; after being incubated for 8 h, observations of their germination status conducted using a 400× optical microscope (Leica Microsystem Inc., Wetzlar, Germany).

#### 2.3.2. Pathogenicity Assessment of *P. italicum*


Gannan navel oranges were selected individually from local orchards, and fruits with similar growth statuses were chosen. Before inoculation, 15 fruits were submerged in a 0.2% (*v*/*v*) sodium hypochlorite solution for 2 min and then washed with sterile water. After air drying, three evenly spaced wounds with a 5 mm diameter were punched at the equatorial region of the fruit and inoculated with 10 μL of spore suspensions (1 × 10^6^ spores/mL), and sterile water was used as the control. Then, the inoculated fruits were incubated at 26 °C for 5 d. The width and length of the areas of decay were measured, and their average was expressed as the diameter of the decay lesion. 

#### 2.3.3. Effects of Various Stress Environments on *P. italicum*

Mycelial discs were prepared as described above; mycelial discs of 5 mm in diameter, exhibiting uniform growth, were inoculated onto PSA medium enriched with a range of stress-inducing agents, including 1 M NaCl, 1 M KCl, 0.5 M CaCl_2_, 0.7 mM CuCl_2_, 2 mM FeSO_4_, 0.3 M MgSO_4_, 0.2 mg/mL SDS, and 0.3 mg/mL congo red. Additionally, to assess the effects of a simple carbon source, sucrose was added to the medium at concentrations of 15 mM, 50 mM, and 175 mM. These cultures were incubated at 26 °C and monitored regularly for any growth or morphological changes. Standard PSA medium was used as the control. Each treatment was replicated three times. After a 7-day incubation, the diameter of the fungal colony was measured. 

#### 2.3.4. Effect of pH on Mycelium Growth of *P. italicum*

Spores of *P. italicum* were harvested following 5 d of cultivation on PSA medium at 26 °C to achieve a concentration of 1 × 10^6^ spores/mL. Subsequently, 100 μL of the spore suspension was inoculated into 100 mL of Potato Sucrose Broth (PSB) medium adjusted to pH levels of 2, 4, 6, 8, and 10; the general PSB was used as a control and then incubated in gyratory shaker at 140 rpm under 26 ± 2 °C. After 48 h, the harvested mycelia were dried in a vacuum freeze dryer for 48 h, and their dry weight were measured.

### 2.4. Observation of Hyphal Ultrastructure by Transmission Electron Microscopy (TEM)

Each 200 µL of spore suspension (3 × 10^6^ spores mL^−1^) from the *P. italicum* was evenly spread onto PSA medium. Then, a sterile coverslip was inserted obliquely into the medium and incubated at 26 °C for 36 h. Additionally, in alignment with protocols established in our preceding studies [33], the hyphae of *P. italicum* were further analyzed utilizing TEM (H-7650, Hitachi, Ltd., Tokyo, Japan).

### 2.5. Cell Death Assay

Employing a modified method from Tao et al., 200 µL of 1 × 10^6^ spores/mL spore suspension was inoculated into 200 mL of sterile PSB and incubated at 26 ± 2 °C, 150 rpm, for 48 h. The mycelium was washed with 0.1 mol/L PBS (pH 7.2) three times, transferred to a slide, and stained with 200 µL of 0.5% Evans blue for 10 min at room temperature. After rinsing off the excess dye with distilled water, mycelium staining was observed under an optical microscope [34].

### 2.6. RNA Transcriptome Sequencing Analysis

Transcriptome analysis was conducted on navel oranges wounded in the equatorial region, similar to the procedure outlined in Section 2.3.4. A quantity of 10 µL of 1 × 10^6^ spores/mL of the control strain *PiKu70* and the knockout strain *ΔPiSntB* was inoculated in five fruits after incubation at 26 °C for 76 h, and tissue at the diseased–healthy interface was harvested, flash-frozen in liquid nitrogen, and stored at −80 °C. 

The sequencing analysis was outsourced to MeiJi BioTech (Shanghai MeiJi BioTech Co., Ltd., Shanghai, China). The total RNA was extracted and sequenced using the Illumina NovaSeq 6000 system, following mRNA isolation and cDNA synthesis. The sequencing involved magnetic beads with Oligo(dT) for mRNA pairing, mRNA fragmentation, cDNA synthesis, Y-shaped adaptor ligation, and amplification. High-quality data were obtained postfiltering with fastp, and alignment to the reference genome (GCA_000769765.1) was achieved using HiSat2. RSEM quantified gene and transcript expression, and DESeq2 identified significant DEGs with FDR < 0.05 and |log_2_FC| ≥ 0.5. Annotation and pathway analysis were conducted across six databases (NR, Swiss-prot, Pfam, COG, GO, and KEGG), providing comprehensive insights into the transcriptome.

### 2.7. Quantitative Real-Time PCR (RT-qPCR) Verification

RT-qPCR was used to verify the relationship between the genes of interest and the RNA-Seq results. RNA was extracted from strains *PiKu70* and *ΔPiSntB*. cDNA synthesis was conducted using the SYBR Premix ExTaq (Tli RNase H Plus) kit (Takara, Japan), adhering to the provided guidelines. RT-qPCR was performed on qTOWER 2.2 (Analytik Jena AG, Germany). Primers (Appendix A) were crafted utilizing Primer Premier 5.0 software (Premier Biosoft International, Palo Alto, CA, USA) and were produced by Wuhan Tianyi Huiyuan Biotechnology Co., Ltd. (Wuhan, China); β-actin served as the reference gene. Gene expression levels were quantified using the 2^−ΔΔCt^ method. The RNA-seq raw datasets were deposited in Sequence Read Archive (SRA), the accession number of which is PRJNA1100932.

### 2.8. Statistic Analysis

Data were presented as mean ± SD, calculated from three independent replicates. Each treatment included three replicates, and experiments were conducted in triplicate. Statistical analyses to identify significant differences between means were performed using one-way ANOVA, with post hoc comparisons conducted using Duncan’s multiple range test. A *p*-value of <0.05 was considered statistically significant. 

## 3. Results

### 3.1. Validation of Mutant Strains

Following the Agrobacterium-mediated gene knockout procedure in *P. italicum*, a mutant strain capable of growing on hygromycin B plates was identified. This transformant was verified using specific primer pairs (Appendix A) Hyg-F/Hyg-R and SntB-F/SntB-R, which confirmed the presence of the hygromycin resistance gene and the absence of ORF amplification of *SntB*. Furthermore, the precise replacement of the *PiSntB* gene with the hygromycin resistance gene was validated using primers SntB-Up-F/Hyg-R and Hyg-F/SntB-Up-R (Appendix A). The knockout strain *ΔPiSntB* was employed in subsequent experiments, with *PiKu70* serving as the control strain.

### 3.2. Effect of SntB Knockout on the Growth and Pathogenicity of P. italicum

To assess the impacts of the *SntB* gene on *P. italicum* growth, we compared the colony morphology, mycelial growth status, spore germination rate, and spore production between the control strain *PiKu70* and the knockout strain *ΔPiSntB*. The results showed that the *ΔPiSntB* strain displayed mycelial growth with a smaller colony diameter in comparison with the *PiKu70* strain, which was reduced by 49.5% (Figure 1A,B). Furthermore, spore germination in the knockout strain was notably delayed, after 9 h of incubation in 0.5% PSA medium, with germination rates of 74% for *PiKu70* and 64% for *ΔPiSntB*. The *ΔPiSntB* strain also showed a 40% reduction in spore production, yielding 1.5 × 10^6^ spores under the same cultivation conditions, compared with 2.5 × 10^6^ spores yielded in the control (Figure 1C). 

The effect of the *ΔPiSntB* mutation on pathogenicity was assessed through artificial inoculation experiments on Gannan navel oranges. The diameter of the decay lesion was reduced by 51.1% on the fifth day postinoculation with *ΔPiSntB* compared with *PiKu70* inoculations (Figure 1D,E), and fewer blue spores and only a small number of white hyphae were observed at the *ΔPiSntB* inoculation sites. Additionally, the inoculation sites with *ΔPiSntB* exhibited much darkening of the lesions, in contrast to those caused by *PiKu70*. These results indicate that the deletion of the *SntB* gene significantly impacted the pathogenicity of *P. italicum*.

### 3.3. Effects of Stress Environments on the Growths of P. italicum

The pH is an important factor that influences the mycelia growth of fungi. As shown in Figure 2A, neither strain was capable of growth at pH 2, indicating intolerance to highly acidic conditions; this was especially true for the knockout strains, which exhibited increased sensitivity to changes in the pH. In mildly acidic conditions (pH 4 and 6), there was a significant reduction in mycelial mass for the knockout strains, with decreases of 52% and 64%, respectively, when compared with the *PiKu70* strain.

To investigate whether *PiSntB* altered the efficiency in utilizing sucrose of the pathogen, different contents of sucrose were added in the culture media. The results demonstrated a positive correlation between the sucrose concentration and the colony radial diameter of the *PiKu70* strain, with a 64% increase in diameter observed with sucrose at 175 mM compared with sucrose at 15 mM. While the *ΔPiSntB* strain showed no significant variation in colony diameter across the different sucrose levels and seemed to grow better under low sucrose levels, relative to the *PiKu70* strain, the *ΔPiSntB* strain exhibited reductions in colony diameter by 59.4%, under high -sucrose conditions (Figure 2B). These findings showed that there was no improvement in growth due to an increase in sucrose concentration.

To examine the impact of *SntB* on *P. italicum* growth under conditions of metal ion stress and cell wall inhibitor presence, both *PiKu70* and *ΔPiSntB* strains were exposed to varying stress levels for seven days, followed by a growth assessment (Figure 2C,D). The results revealed significant inhibition of mycelial growth of the strains in culture media rich in metal ions such as K^+^, Na^+^, Ca^2+^, Cu^2+^, and cell wall disruptors. Fe^2+^ and Mg^2+^ ions positively influenced growth, enhancing the *PiKu70* colony diameter by 12.7% and 30.9%, respectively. The *ΔPiSntB* mutant strain exhibited increased sensitivity to all treatments compared with *PiKu70*. Intriguingly, the Fe^2+^ ion, rather than facilitating growth, led to a 7.1% decrease in the colony diameter for *ΔPiSntB*. These findings implied that the deletion of the *SntB* gene significantly affected *P. italicum*’s adaptability to various cationic and stress conditions and reduced the normal utilization of iron ions.

### 3.4. Effects of SntB Knockout on the Ultrastructure of Hypahe

As demonstrated in Figure 3A,B, organelles within the control group exhibited normal distribution and morphology, characterized by being well defined in size and shape, indicating that the cells were in a healthy state. In contrast, hyphae from the *ΔPiSntB* strain revealed significant cellular anomalies, as detailed in Figure 3C,D. These anomalies included pronounced cell wall thickening, cell membrane invaginations, disorganized cytoplasm, and an increase in autophagic vesicles. There was evident nuclear membrane breakage and leakage of nucleoplasm, suggesting compromised cellular integrity. Additionally, mitochondria not only expanded but also increased in number, signifying that the absence of *SntB* gene expression may have initiated autophagy-like and stress-responsive pathways, ultimately leading to the destruction of cellular structures in *P. italicum*.

Further supporting these findings, hyphae from the *ΔPiSntB* strains displayed positive staining with Evans blue (Appendix A), indicating compromised cell wall integrity, which contrasted with the normal appearance of the *PiKu70* strain hyphae. This observation underscored the role of the *SntB* gene in maintaining cell structure integrity, with its deletion rendering the mycelia more susceptible to damage.

### 3.5. Transcriptomic Analysis of Differentially Expressed Genes in Knockout Strains

Transcriptome sequencing was employed to uncover the molecular mechanisms through which the deletion of the *SntB* gene reduced the host pathogenicity of *P. italicum*. The analysis revealed that 2507 upregulated and 1976 downregulated genes were observed in comparison with the *PiKu70* strain (Figure 4A). 

Gene Ontology (GO) enrichment analysis categorized the differentially expressed genes into three primary categories: cellular components, biological processes, and molecular functions. The top 20 GO terms (Figure 4B) highlighted significant enrichment in molecular functions such as catalytic activity, cell membrane composition, and hydrolase activity. In cellular components, membrane transporter and oxidoreductase activities were notably enriched. Biological processes showed significant enrichment in carbohydrate metabolic processes.

The Kyoto Encyclopedia of Genes and Genomes (KEGG) pathway analysis revealed that a total of 62 pathways were significantly enriched, including carbohydrate metabolism (starch and sucrose metabolism, glycolysis, and pyruvate metabolism), amino acid metabolism (arginine, proline, tryptophan, and serine), and lipid metabolism (fatty acid degradation and glycerolipid metabolism). Furthermore, pathways associated with stress tolerance and secondary metabolite biosynthesis were also enriched (Figure 4C). RT-qPCR validation confirmed the reliability of the RNA-seq results.

#### 3.5.1. Deletion of SntB Leads to Differential Expressed Genes Related to Carbohydrate Metabolism

Carbohydrate metabolism is essential for fungal growth, not only providing energy but also sustaining fungal survival and reproduction by regulating physiological processes and adapting to environmental alternations. More than 40 upregulation genes were observed in hydrolase, such as alpha-amylase (PITC_008490), carbon-nitrogen hydrolase (PITC_069100 and PITC_029870), glucose-repressible protein Grg1 (PITC_045240 and PITC_061580), phosphoenolpyruvate carboxykinase, ATP-utilizing (PITC_095870), and the glycoside hydrolase family (PITC_058850 and PITC_003820). However, in the carbohydrate metabolism pathway, 6-phosphofructokinase (PITC_097960), enolase (PITC_089720), and phosphoglycerate kinase (PITC_041290), which are involved in glycolysis, were downregulated. This suggested a reduction in ATP synthesis, and all of these genes were expressed in high FPKM values.

#### 3.5.2. Deletion of SntB Leads to Expression Changes in Lipid Metabolism-Related Genes

Lipid metabolism serves as one of the crucial energy sources for cells and is essential for the integrity and functionality of biological membranes [35]. It was observed that there was an upregulation of genes implicated in lipid hydrolysis, specifically lipase (PITC_051810 and PITC_083320) and alpha/beta hydrolase (PITC_032560). Additionally, a significant upregulation was noted in more than 30 genes encoding short-chain dehydrogenase/reductase (SDR) enzymes (PITC_038780 and PITC_052730), highlighting an enhanced lipid metabolism process. Conversely, genes associated with lipid biosynthesis, such as acyl-CoA N-acyltransferase (PITC_020530), fatty acid desaturase (PITC_098660, PITC_031890), acyl transferase/acyl hydrolase/lysophospholipase (PITC_023140, PITC_090390), and phospholipid methyltransferase (PITC_013770), were observed to be significantly downregulated. This result suggested a coordinated regulation of lipid metabolism favoring hydrolysis over biosynthesis under the studied conditions.

#### 3.5.3. Deletion of SntB Leads to Higher Demand for Energy Metabolism

We observed that mitochondrial carrier proteins (PITC_016450, PITC_088820, and PITC_095900), NADH-ubiquinone oxidoreductase (PITC_081320 and PITC_023670), class cytochromes (PITC_071230, PITC_075670, PITC_070840, PITC_069900, PITC_083400, and PITC_056470), phosphoesterase (PITC_014370 and PITC_088100), and FAD-dependent pyridine nucleotide-disulfide oxidoreductase (PITC_058760) with high expression values were upregulated, which implied a high energy demand by the organism, and the degree of oxidative phosphorylation was deepened. Concomitantly, the downregulation of mitochondrial inner and outer membrane proteins (PITC_030560 and PITC_066990) indicated a coordinated effort to reduce energy expenditure. This reorganization of mitochondrial membrane structure reflected an adaptive response aimed at enhancing cellular functionality. 

#### 3.5.4. Deletion of SntB Altered the Expressions of Genes Involved in Cellular Structure and Material Transport

Transporter proteins, as a class of biomolecules, allow specific types of molecules or ions to cross cell membranes and facilitate the uptake, excretion, and distribution of substances [36]. There was a pronounced upregulation of dynamin proteins (PITC_083340 and PITC_057950) and members of the major facilitator superfamily (MFS) transporters (PITC_058840 and PITC_005580), which was more than 150 genes documented in this study. Among these, MFS transporters displayed a remarkable upregulation with high log_2_ fold changes (log_2_FC > 4). Concurrently, a downregulation of several transporter genes was noted, including those coding for amino acid transporter (PITC_083180), vacuolar protein (PITC_040980), permease (PITC_033280), malic acid transport protein (PITC_006380), and the sodium/calcium exchanger membrane region (PITC_066290). Additionally, an upsurge in the expression of genes involved in cell wall beta-glucan synthesis (PITC_008500), the glucanases superfamily (PITC_002540 and PITC_041170), concanavalin A-like lectin/glucanases superfamily (PITC_002540 and PITC_041170), and hydrophobin (PITC_015600, PITC_001010, and PITC_077120) were detected. Conversely, genes encoding cytoskeletal elements such as actin (PITC_071360) and tubulin (PITC_003900) were downregulated. This transcriptional shift suggested an ongoing remodeling of the cytoskeleton and cell wall (Appendix A).

#### 3.5.5. Deletion of SntB Resulted in Significant Changes in Amino Acid/Protein Metabolism

The balance of amino acid/protein metabolism is essential for sustaining growth in fungal cells. Amidases (PITC_001990 and PITC_079890) and D-amino-acid oxidase (PITC_029880) were significantly upregulated, alongside copper amine oxidases (PITC_025260 and PITC_072940), which exhibited exceptionally high expression with FPKM values of 2292 and 1256, respectively. Concurrently, there was an upregulation of serine hydrolases (PITC_039780 and PITC_008170) and peptidases (PITC_025820 and PITC_045460), indicating that proteolysis was enhanced. In contrast, ribosomal proteins (PITC_073660 and PITC_089630) were downregulated, suggesting a reduction in the cellular capacity for protein synthesis (Appendix A). This pattern highlighted a metabolic shift favoring amino acid degradation over protein synthesis in the cellular context.

#### 3.5.6. Deletion of the SntB Gene Triggers a Series of Oxidative Stress and Immune Response

Fungi navigate the challenges posed by both endogenous and exogenous oxidants through the activation of specific enzymes that contribute to maintaining redox equilibrium and facilitating detoxification [37,38,39]. The significant upregulation of cytochrome P450 (PITC_052690 and PITC_005950) with a log_2_ FC > 8, serine-threonine protein kinase (PITC_037590), cupin protein (PITC_027250), glutathione S-transferase (PITC_033490, PITC_031710, PITC_012420, and PITC_003880), and aldehyde dehydrogenase (PITC_027470, PITC_032430) was observed, indicating that in response to oxidative stress, cells regulated the intracellular redox balance, aiming to eliminate excess metabolic waste. Additionally, the upregulation of oxoglutarate/iron-dependent dioxygenases (PITC_000490, PITC_024150, and PITC_030020) with a log_2_ FC = 5.66 suggested that the organism was responding to complex environmental stimuli, including the perception of iron homeostasis. However, the downregulation of electron transfer flavoprotein-ubiquinone oxidoreductase (PITC_060320), alcohol dehydrogenase (PITC_097450), Clp ATPase (PITC_040670), and HSP20-like chaperone (PITC_090090) indicated that the damage caused by oxidative stress within the organism had not been significantly mitigated.

#### 3.5.7. Deletion of the SntB Gene Downregulated Pathogenicity-Related Genes

The cell wall degrading enzyme (CDWE) is one of the most important virulence factors, the CDWE-related DEGs including pectin lyase (PITC_034160) and cutinase (PITC_030230) were significantly downregulated, and the protein aegerolysin protein (PITC_014830) related to cell lysis and conidiation was also significantly downregulated, which may have affected its ability to infect its host.

### 3.6. Gene Expression Measurements

Upon evaluating the expression of pivotal regulators involved in glucose-repressible protein Grg1, oxoglutarate/iron-dependent dioxygenase, short-chain dehydrogenase/reductase SDR, Actin-binding, cofilin/tropomyosin type, and NADH-ubiquinone oxidoreductase via qRT-PCR, we noted the expression trend was consistent with the transcriptome data, which could prove the accuracy of the transcriptome data. At the same time, the observed downregulation of key regulatory genes involved in secondary metabolism (*LaeA*) and pH regulation (*PacC*) and upregulation of carbon metabolism inhibition (*CreA*) following the deletion of the *SntB* gene highlighted its vital role in modulating these essential pathways (Figure 5). 

## 4. Discussion

This study successfully created *SntB* gene knockout mutants in *P. italicum*, leading to significant phenotypic changes. Notably, the mutants exhibited inhibited vegetative growth, reduced sporulation, and decreased pathogenicity on citrus fruits, a phenomenon consistent with observations in *SntB* knockout strains of *A. flavus*, *Fusarium oxysporum*, and *Neurospora crassa*. Similar phenomena were observed [11,12,40]. The role of mycotoxins in host pathogenicity has been explored in other fungi [41,42]; however, in *P. italicum*, no mycotoxins have been reported in decayed fruits to date, nor have the secondary metabolites involved in virulence been identified. In our previous experiments, we tried to detect potential toxins (tryptoquialanines and citrinin) of citrus pathogen [43], but none were detected; hence, mycotoxins will not be further discussed in this manuscript.

Subsequent analyses of *SntB*’s role in *P. italicum* revealed a diminished ability to assimilate simple carbon sources like sucrose, impaired organelle functionality, and increased sensitivity to environmental stress in knockout strains compared with the parental strain. The transcription factor *CreA*, which mediates carbon catabolite repression (CCR), a glucose-mediated repression mechanism, is typically involved in prioritizing glucose utilization over other carbon sources [44,45,46]. In the *ΔPiSntB* strain, CCR was disrupted, as evidenced by the inability to mitigate growth impairment with increased sucrose concentrations, in contrast to the parental *PiKu70* strain. This disruption hindered normal carbon source utilization, leading to interference in the CCR pathway, impacting sugar metabolism, and ultimately resulting in growth and developmental delays [46]. Intriguingly, environmental factors and host interactions appeared to influence fungal gene expression, as indicated by a slight upregulation of the *creA* inoculated with navel orange. For example, in *Alternaria citri*, *AcCreA* upregulation relieved CCR, enabling the synthesis of endopolygalacturonase, a key virulence factor, and consequently reducing pathogenicity in fruit [47].

The inability to efficiently utilize sucrose suggested carbon starvation in the fungi, a state where available carbon sources were insufficient to support vegetative growth, prompting a stress response to survive [48]. For instance, the metabolic balance shifted toward degradation, as evidenced by the induction of macroautophagy and increased production of hydrolytic enzymes, allowing nutrient liberation through the breakdown of cellular constituents for prolonged survival [49]. The experimental results suggested that the *SntB* gene deletion perturbed the CCR pathway, inducing carbon starvation and consequent metabolic pathway alterations, as evidenced by transcriptomic analysis. Notably, there was a marked upregulation of hydrolases, such as alpha-amylase, carbon-nitrogen hydrolase, and glycoside hydrolase, implicated in carbohydrate metabolism. Furthermore, amidase and peptidase related to protein synthesis and degradation, along with alpha/beta hydrolase and lipase linked to lipid metabolism, were upregulated. The downregulation of synthases, including enolase and amidotransferase, illuminated their crucial role in cellular adaptation to environmental stress and the maintenance of energy balance and nutritional metabolism, paralleling observations in other fungi [50].

Previous studies have shown that carbon starvation induces extracellular hydrolase production, increased reactive oxygen species (ROS), modifications in glutathione metabolism, upregulation of certain antioxidant enzymes, ammonia release, and a reduction in respiration [51]. It has also been demonstrated that the carbon starvation stress response (CSSR) in *A. nidulans* may lead to apoptotic cell death in aging cultures [52]. Our study demonstrated significant upregulation of oxidoreductases, including NADH-ubiquinone oxidoreductase, cytochrome c, and FAD-dependent pyridine nucleotide-disulfide oxidoreductase, all of which are known to contribute to elevated ROS levels [53,54], thus corroborating the occurrence of carbon starvation. This oxidative stress was further evidenced by the upregulation of cytochrome P450 and glutathione S-transferase, suggesting an enhanced role for redox reactions in coping with this starvation-induced stress.

Starvation-induced autolysis and autophagy, crucial for nutrient recycling, were highlighted by our transcriptomic data and were consistent with previous findings. In *A. nidulans*, for instance, carbon starvation precipitates autolysis, characterized by an upregulation of hydrolytic enzymes [45,55]. Autophagy likely plays an integral role in nutrient recycling during carbon starvation, with *A. niger atg* mutants illustrating how autophagy supports mycelial survival under carbon depletion [56]. Moreover, the elevated expression of transport proteins for amino acids and peptides implies the recycling of components released during autolysis to support survival during carbon starvation [57]. Our observation revealed that the *ΔPiSntB* strain was more susceptible to metal ions, particularly with impaired Fe^2+^ utilization. The transcriptomic data underscored the upregulation of Oxoglutarate/iron-dependent dioxygenase. Furthermore, Evans blue staining and transmission electron microscopy suggested compromised hyphal integrity and increased autophagosomes, hinting at ferroptosis due to improper iron utilization, a hypothesis that warrants further investigation.

In conclusion, *PiSntB* deletion disrupts the CCR pathway, triggering carbon starvation, oxidative stress, and autophagy. These events hinder fungal growth, elevate stress susceptibility, damage internal organelles, and may precipitate cell death. Such compromised growth diminishes pathogenicity toward citrus fruits, providing insights into the molecular dynamics of fruit invasion by pathogens and informing potential biocontrol strategies.

## Figures and Tables

**Figure 1 jof-10-00368-f001:**
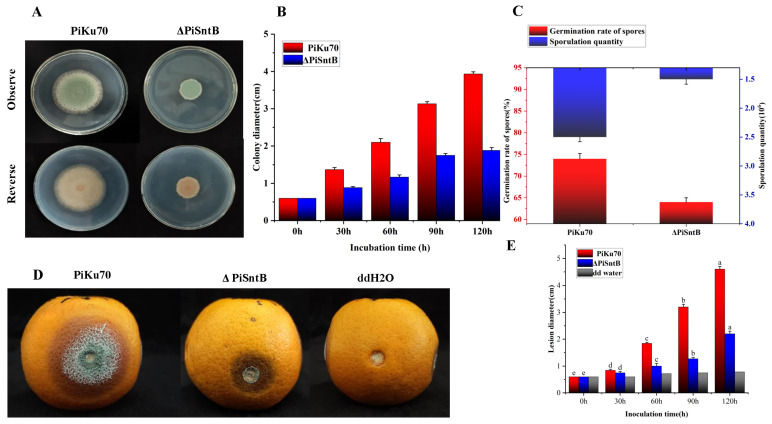
Effects of the *SntB* gene deficiency on the growth and pathogenicity of *P. italicum*. (**A**,**B**) Colony morphology and colony radial growth under solid culture conditions. (**C**) Spore production and spore germination rate. (**D**) Incidence of *P. italicum* of navel oranges at 5d. (**E**) Mean diameters of disease lesions at inoculation sites developed with time. Different letters (a, b, c, d or e) above the columns indicate significant differences between the groups (*p* < 0.05).

**Figure 2 jof-10-00368-f002:**
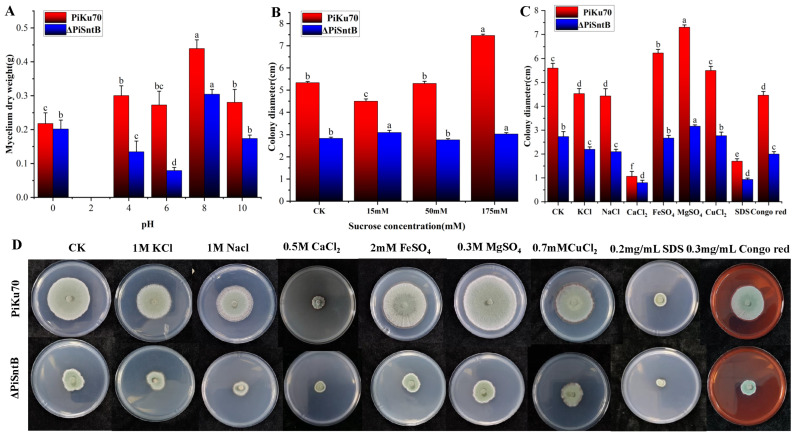
Radial growth of *P. italicum* stains under stress environment. (**A**) Mycelial mass at different pH (2, 4, 6, 8, and 10) conditions, where group 0 is normal PSB medium. (**B**) Colony diameter of two strains at low, medium, and high sucrose concentrations, where the CK group is normal PSA medium. (**C**,**D**) Colony diameter and colony state in media containing metal ion and cell wall disrupters. Different letters (a, b, c, d, e or f) above the columns indicate significant differences between the groups (*p* < 0.05).

**Figure 3 jof-10-00368-f003:**
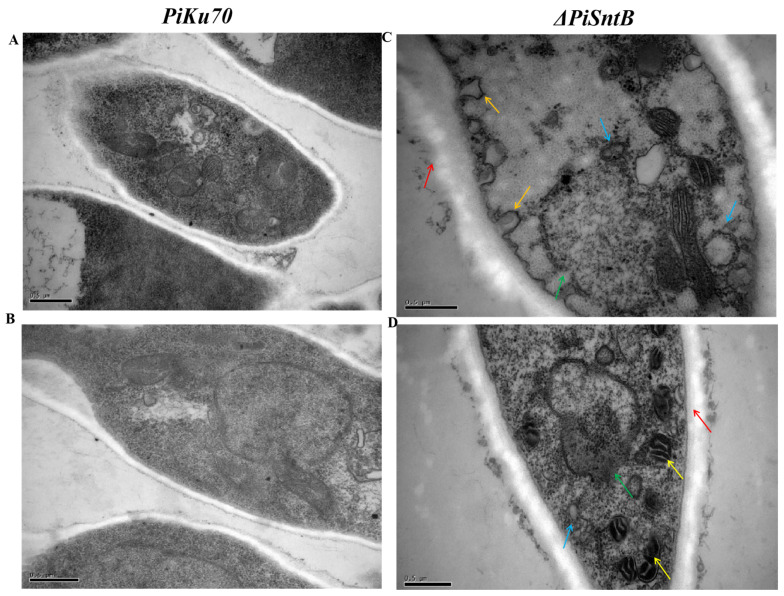
Ultrastructure of *P. italicum* hyphae observed by transmission electron microscopy (TEM). (**A**,**B**) *PiKu70*, (**C**,**D**) of *ΔPiSntB*. Green arrow: degradation of the nuclear membrane and release of nucleoplasm; red arrow: thickening of the cell wall; blue arrow: increase in autophagic vesicles; yellow arrow: mitochondrial expansion; orange arrow: cell membrane invaginations, as if the cell wall skeleton collapsed.

**Figure 4 jof-10-00368-f004:**
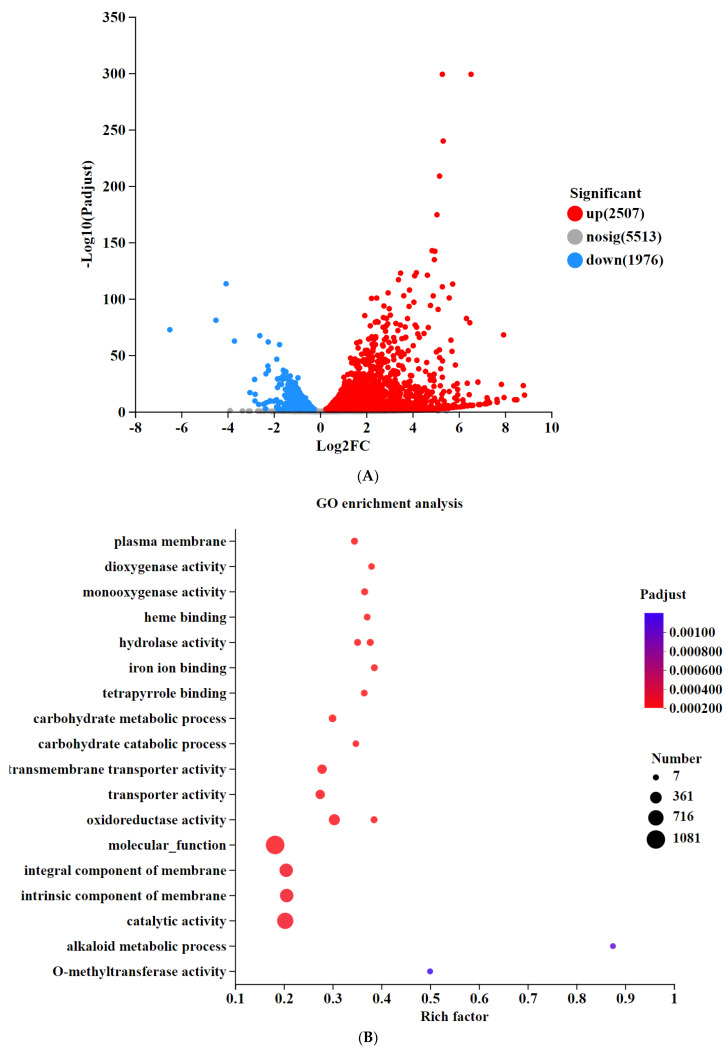
Analysis of differentially expressed genes (DEGs) in *ΔPiSntB. (***A**) Cluster heat map of DEGs in *ΔPiSntB* and *PiKu70*. (**B**): Gene Ontology enrichment analysis of differentially expressed genes in *P. italicum*. (**C**) KEGG enrichment analysis of differentially expressed genes in *P. italicum*.

**Figure 5 jof-10-00368-f005:**
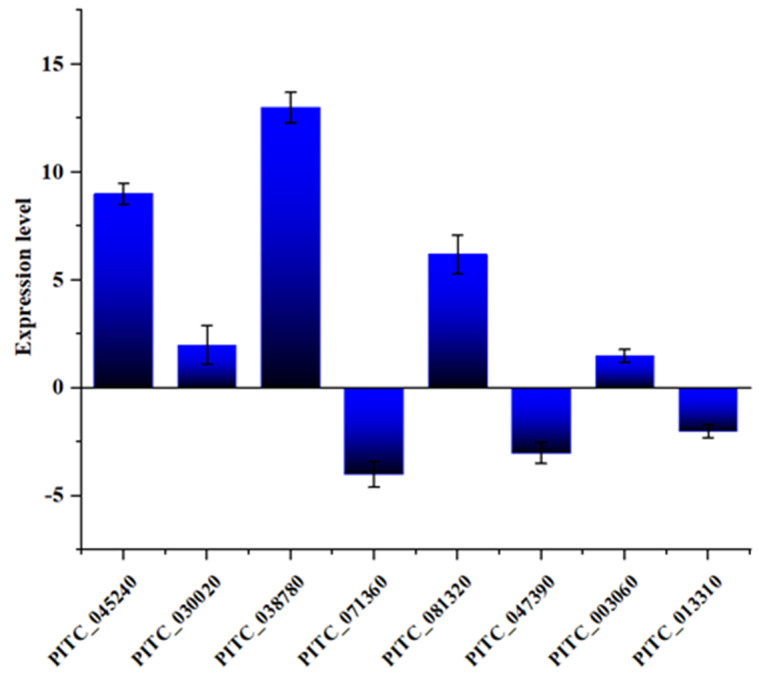
RT-qPCR verification of DEGs in RNA-Seq.

## Data Availability

All the raw RNA-seq data analyzed in this study can be freely downloaded from the SRA database in NCBI, with the sample accession number PRJNA1100932.

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
