# Peer review of "SntB Affects Growth to Regulate Infecting Potential in Penicillium italicum"

_jof, 2024, doi:10.3390/jof10060368_

Round 1

Reviewer 1 Report

The work addresses the aspects in which the SntB gene could be involved, pointing out its role in Carbohydrate Metabolism, Lipid Metabolism-Related Genes, Energy Metabolism, Cellular Structure and Material Transport, Amino Acid/Protein Metabolism, which in short is all types of metabolisms basal that ensures the survival and development of the fungus. Deletion of this gene Triggers a Series of Oxidative Stress and Immune Response and Downregulated Pathogenicity-related Genes. But given its impact at the basal level, this effect seems more like an indirect effect caused by its involvement in the development of the fungus.

-Line 30:  in the introduction it would be necessary to include a quote.

-Lines 49-57: everything known about these three genes should be developed in more depth and specific cases of postharvest pathogens explained.

-Lines 66-67: It would be good to explain the reason for the selection of Ku70 gene deficient strain from P. italicum.

-Line 108-112: it is not clear why the inoculation point is so large, since this wound already provokes a series of responses. And not only the effect on the severity of the damage but also the incidence (number of fruits affected) should be provided.

-Line 113: what is the advantage of placing mycelium discs to test the Effects of Various Stress Environments and why not do it with a spore solution where the effect would be seen not only of the mycelium but also of germination?

-Line 211: which refers to Effects of Stress Environments on the Growths of P. italicum as it is represented, it may lead to confusion since it must be taken into account that the starting point of the deletant is already determined by a notable decrease in its growth and this is reflected in all types of stresses.

- A very detailed reflection is made of all the genes involved in the processes of Carbohydrate Metabolism, Lipid Metabolism-Related Genes, Energy Metabolism, Cellular Structure and Material Transport, Amino Acid/Protein Metabolism, however very little reference is made to genes of pathogenicity, is it that more relevant genes were not identified?

- I would include Figure S3 in the text and Table 1 would be included in supplementary material. The effects at the expression level should be relevant.

Perhaps it would be interesting to highlight that SntB ultimately hinders the growth of fungi, increases susceptibility to stress, damages internal organelles and can precipitate cell death and, as a consequence, affects their infective capacity. Therefore, its role has to do with basal aspects of growth and not pathogenicity and could be indicated in some way in the title to make it more informative.

Author Response

Dear  reviewer,

Thank you all so much for your reviewing this work (Manuscript ID:jof-2999940, Title: The Regulatory Effect of SntB on Penicillium italicum from Postharvest Citrus).

First, we would like to show our respects to all the reviewers, because they are competent, and their comments and suggestions are highly valuable, and help we a lot.

According to their comments, we tried our best to revise the manuscript. And now, the detailed explanations for the revision are listed as following:

Reviewer 1

Comments 1: Line 30: in the introduction it would be necessary to include a quote.

Response 1:

Thank you very much for the constructive suggestions, two references has been added on Page 1, Line 35, as shown in the revised version.

Comments 2: Lines 49-57: everything known about these three genes should be developed in more depth and specific cases of postharvest pathogens explained.

Response 2:

It is really a constructive advice, we rewrite this part about the three genes in depth and specific cases of pathgens explaned, as shown in the revised version.

Other notable virulence factors include LaeA, PacC, CreA. LaeA is regarded as the secondary metabolism global regulator, since it coordinates the expression of various secondary metabolite clusters [13-15]. In P. expansum, LaeA regulates the biosynthesis of patulin, which impacts virulence related to this pathogen and is mediated by sucrose [16]; The pH regulatory factor PacC, promotes pathogen colonization by generating organic acids to reduce host pH at infection sites [17,18]. In both P. expansum and Aspergillus carbonarius, PacC directly regulates glucose oxidase (GOX), which catalyzes the oxidation of glucose to gluconic acid, facilitating host colonization [19,20]. In Fusarium fujikuroi, ΔpacC mutants did not exhibit expression of FUB1, the key polyketide synthase gene in the fusaric acid gene cluster [21,22]; CreA is the carbon metabolism repression regulator, and regulates carbon source utilization by modulating carbon catabolite repression (CCR) activity in filamentous fungi [23]. Recent studies have discovered that CreA regulates growth and pathogenicity in Valsa mali, and that MoCreA is essential for asexual development and pathogenicity in Magnaporthe oryzae [24,25]. Notably, SntB has been reported to positively regulate these global virulence and secondary metabolism regulators [10], suggesting its involvement in multiple critical biological processes and metabolic pathways. According to your suggestion, the description has been added on Page2, Line 53-69.

Comments 3: Lines 66-67: It would be good to explain the reason for the selection of Ku70 gene deficient strain from P. italicum.

Response 3:

Thank you for pointing this out. To enhance the efficiency of homologous recombination, we developed a Ku70-deficient strain from P. italicum, this strain was employed as the parental control for all further experimental activities. According to your suggestion, the description has been added on Page 2, Line 78-80.

Comments 4: Line 108-112: it is not clear why the inoculation point is so large, since this wound already provokes a series of responses. And not only the effect on the severity of the damage but also the incidence (number of fruits affected) should be provided.

Response 4:

Thank you very much for the constructive suggestions. Before inoculation, 15 fruits were submerged in a 0.2 % (v/v) sodium hypochlorite solution for 2 min and then washed with sterile water. According to your suggestion, the description has been added on Page 3, Line 120.

In response to the query regarding the size of the inoculation sites, we would like to clarify that we used a 5mm punch to ensure uniform initial diameters across all inoculation sites. This approach not only preserved the depth of each hole without damaging the fruit flesh but also enhanced the reproducibility of the operations. The circular shape of the inoculation sites further contributed to the regularity of the disease lesions. It is important to note that during the observation period, inoculation sites injected with sterile water did not exhibit any rot, indicating that the punching method did not adversely affect the experimental outcomes. Additionally, my primary research focus is on P. italicum. The subsequent transcriptomic data primarily explored changes in the pathogen's genes, without studying alterations in the citrus genes. The inoculation sites on the navel oranges did not significantly affect the state of the pathogen itself.

Comments 5: Line 113: what is the advantage of placing mycelium discs to test the Effects of Various Stress Environments and why not do it with a spore solution where the effect would be seen not only of the mycelium but also of germination?

Response 5:

It is really a good question. Using mycelium discs to test different stress environments is a standard method in our laboratory.we would like to highlight several advantages of this approach: the mycelium discs are consistently 5mm in diameter throughout the experiment, which not only allows for direct observation of whether the colonies grow regularly but also enables us to measure colony diameters to assess growth rates. This method also permits direct observation of mycelial and spore colors, ensuring the rigor of the experiment.

While using a spore solution for testing is a viable alternative that allows for the observation of mycelium and germination, accurately counting spore numbers can be challenging. Additionally, mycelium grown in liquid culture may exhibit slight variations due to differences in rotating speed, potentially introducing experimental errors, which is why this method was not exclusively used.

Furthermore, in this experiment, we employed Scanning Electron Microscopy (SEM) to observe the surface state of the hypahe. Since there were no significant differences compared to the control group, these findings were not presented in the paper.

Comments 6: Line 211: which refers to Effects of Stress Environments on the Growths of P. italicum as it is represented, it may lead to confusion since it must be taken into account that the starting point of the deletant is already determined by a notable decrease in its growth and this is reflected in all types of stresses.

- A very detailed reflection is made of all the genes involved in the processes of Carbohydrate Metabolism, Lipid Metabolism-Related Genes, Energy Metabolism, Cellular Structure and Material Transport, Amino Acid/Protein Metabolism, however very little reference is made to genes of pathogenicity, is it that more relevant genes were not identified?

- I would include Figure S3 in the text and Table 1 would be included in supplementary material. The effects at the expression level should be relevant.

Perhaps it would be interesting to highlight that SntB ultimately hinders the growth of fungi, increases susceptibility to stress, damages internal organelles and can precipitate cell death and, as a consequence, affects their infective capacity. Therefore, its role has to do with basal aspects of growth and not pathogenicity and could be indicated in some way in the title to make it more informative.

Response 6:

Thank you very much for your constructive suggestions. You are absolutely correct—the experimental results of this paper indeed indicate that the absence of SntB disrupts normal growth of fungi, leading to abnormal phenotypes, which reduce the infectivity of the strain on citrus. To avoid any ambiguity, we have revised the title to "SntB Affects Growth to Regulate Infecting Potential in Penicillium italicum", which aligns more closely with the main theme of the manuscript.

Furthermore, following your advice, we will move Figure S3 into Section 3.6 under Gene Expression Measurements and rename it as Figure 5. It has been added on Page 12, Line 410-413.

Additionally, we will relocate Table 1 to the supplementary materials, where it will be designated as Table S3. However, if possible, we would prefer to keep Table 1 in the main text to provide readers with easier access to the descriptions of genes and their expression levels.

Once again, we are greatly appreciated of your critical reviews and valuable comments and suggesttions. If you have any other questions about the revised version of this manuscript, please do not hesitate to let us know.

Yours Sincerely,

Chunyan Li

Reviewer 2 Report

This study successfully created SntB gene knockout mutants in P. italicum, leading to  significant phenotypic changes. The autors reports that the mutants exhibited inhibited vegetative growth, reduced sporulation, and decreased pathogenicity on citrus fruits. Ultrastructural analyses disclosed notable disruptions in cell membrane integrity, disorganization within the cellular  matrix, and signs of autophagy. Transcriptomic data further indicates a pronounced upregulation  of hydrolytic enzymes, oxidoreductases, and transport proteins, suggesting a heightened energy  demand. The observed phenomena are consistent with a carbon starvation response potentially triggering apoptotic pathways, including iron-dependent cell death. These findings collectively under score the pivotal role of SntB in maintaining the pathogenic traits of P. italicum, proposing that targeting PiSntB could offer a new avenue for controlling citrus fungal infections and subsequent fruit decay.

This manuscript is relevant and an important contribution to know the pathogenicity pathway of postharvest pathogens.

I have made a few corrections in a copy of the pdf manuscript.

In general, the figures need to increase the letter sizes because it is difficult to read.

Author Response

Dear reviewer, 

Thank you all so much for your reviewing this work (Manuscript ID:jof-2999940, Title: The Regulatory Effect of SntB on Penicillium italicum from Postharvest Citrus).

First, we would like to show our respects to all the reviewers, because they are competent, and their comments and suggestions are highly valuable, and help we a lot.

According to their comments, we tried our best to revise the manuscript. And now, the detailed explanations for the revision are listed as following:

Basic suggestions : In general, the figures need to increase the letter sizes because it is difficult to read.

Response:

Thank you very much for your constructive suggestions. In accordance with your suggestion, we have increased the font size in the figures and enhanced the resolution of the images to ensure clear readability.

Comments 1:Line 26: 30-50% This percentages is very high. This need revision. Please check. "The most harmful phytopathogenic fungi of oranges are Penicillium digitatum, which causes the green mold disease, responsible for about 90% of post-harvest losses (Costa et al., 2019b; Papoutsis et al., 2019), and Penicillium italicum Wehmer, the causing agent of the blue mold disease. The latter disease develops more slowly, however, it presents higher resistance to cold (Whiteside et al., 1993; Palou et al., 2002; Iqbal et al., 2012, 2017) and to low water availability (Plaza et al., 2003), easily spreading and contaminating a greater number of healthy oranges. The presence of wounds in the fruit surface is essential for infection by these fungi (Caccioni et al., 1998; Talibi et al., 2014)."

Response 1:

Thank you very much for your constructive suggestions. Indeed, the information previously presented was inaccurate. we have revised the sentence accordingly to:

Among the various post-harvest diseases affecting citrus, green and blue mold stand out as significant industrial challenges [2,3]. Notably, blue mold, which is induced by Penicillium italicum, progresses more gradually but exhibits substantial resistance to cold conditions, low water availability, and has a propensity for rapid spread and contamination of numerous healthy oranges [4-6], This can lead to considerable economic damages. According to your suggestion, the description has been added on Page 1, Line 26-31.

Comments 2: Figure. 1E The gray bars msust be zero, because it is the control values and the control was not inoculated with conidias

Response 2:

Thank you very much for your constructive suggestions. we have revised the figure as per your suggestions.

Once again, we are greatly appreciated of your critical reviews and valuable comments and suggesttions. If you have any other questions about the revised version of this manuscript, please do not hesitate to let us know.

Yours Sincerely,

Chunyan Li

Round 2

Reviewer 1 Report

The authors have followed the suggestions point by point, responding and changing the sections that were not clear enough. With this they have managed to improve the quality of the manuscript so that it is accepted in its current form.

-The explanations for the method of infection are debatable, because the wound performed does affect the response.

-Stress treatment using mycelium can be used, but inoculation with spores is sometimes more informative and perfectly reproducible.